# Optical vortex-antivortex crystallization in free space

Haolin Lin[1,2,6], Yixuan Liao[1,2,6], Guohua Liu [1,2], Jianbin Ren[1,2], Zhen Li[1,2,3] ✉, Zhenqiang Chen[1,2,3] ✉, Boris A. Malomed [4,5] & Shenhe Fu [1,2,3] ✉

Stable vortex lattices are basic dynamical patterns which have been demonstrated in physical systems including superconductor physics, Bose-Einstein condensates, hydrodynamics and optics. Vortex-antivortex (VAV) ensembles can be produced, self-organizing into the respective polar lattices. However, these structures are in general highly unstable due to the strong VAV attraction. Here, we demonstrate that multiple optical VAV clusters nested in the propagating coherent field can crystallize into patterns which preserve their lattice structures over distance up to several Rayleigh lengths. To explain this phenomenon, we present a model for effective interactions between the vortices and antivortices at different lattice sites. The observed VAV crystallization is a consequence of the globally balanced VAV couplings. As the crystallization does not require the presence of nonlinearities and appears in free space, it may find applications to high-capacity optical communications and multiparticle manipulations. Our findings suggest possibilities for constructing VAV complexes through the orbit-orbit couplings, which differs from the extensively studied spin-orbit couplings.

Optical vortices in their basic form are represented by topological solutions of the paraxial Helmholtz equation. They are distinguished by the helical phase factor $\exp(il\phi)$, combined with either the Laguerre-Gaussian or Bessel-Gaussian amplitude profiles, where $\phi$ is the azimuthal coordinate, and integer $l$ is the topological charge (alias winding number). The vortex beam exhibits a phase dislocation at the vortex pivot and carries a well-defined intrinsic orbital angular momentum (OAM)[1], which has found various applications, in the classical and quantum regimes alike[2–5]. For example, by appropriately introducing multiple vortices with different topological charges into a single beam, one can considerably enhance the optical communication capacity and speed[6–8].

Optical vortex-antivortex (VAV) lattices, which carry OAM with positive and negative signs, i.e., opposite topological charges, are a fundamentally important concept that can find promising applications,

including high-capacity optical communications[6–8], parallelized superresolution[9], multiparticle manipulations[10–12], and higher-dimensional quantum information processing[13–15]. In the past, significant advancements have been made on the techniques of generating vortex arrays in the linear regime. For instance, the vortex arrays with various designs can be created by coaxially superimposing different fundamental modes with appropriate weighting coefficients, such as Laguerre-Gaussian[16,17], Hermite-Gaussian[16], Ince-Gaussian[18], Bessel-Gaussian[19], perfect optical vortex[20,21] and other vortex fields with curvilinear shapes[22]. The large-scale vortex lattices are produced by the far-field interferences of planar waves[23–25], or vortex fields[26]. Another simple way is to arrange multiple non-coaxial unit cells, including Laguerre-Gaussian or perfect-optical-vortex, on interstitial sites[27–33]. However, the propagation dynamics of these VAV arrays/lattices induced by couplings between the vortex-antivortex pivots both in linear[34–37] and nonlinear

[1]Department of Optoelectronic Engineering, Jinan University, Guangzhou 510632, China. [2]Guangdong Provincial Key Laboratory of Optical Fiber Sensing and Communications, Guangzhou 510632, China. [3]Guangdong Provincial Engineering Research Center of Crystal and Laser Technology, Guangzhou 510632, China. [4]Department of Physical Electronics, Faculty of Engineering, Tel Aviv University, Tel Aviv 69978, Israel. [5]Instituto de Alta Investigación, Universidad de Tarapacá, Casilla 7D, Arica, Chile. [6]These authors contributed equally: Haolin Lin and Yixuan Liao. ✉e-mail: ailz268@126.com; tzqchen@jnu.edu.cn; fushenhe@jnu.edu.cn

media[38] was not systematically studied. The presence of nonlocal couplings between phase dislocations in the VAV wavefront makes them strongly unstable against initial perturbations of the lattice structure[39]. Specifically, lattices composed of homopolar vortices perform perturbation-induced rotation in the course of the propagation, as a direct consequence of the straight motion of vortices in the transverse plane[40,41]. Furthermore, VAV pairs in the lattices demonstrate annihilation, repulsion[36,39,40,42], or re-creation (the intrinsic OAM Hall effect)[37,39]. The instability ensuing from diverse coupling processes between vortices and antivortices transforms initially regular lattices into quasi- or totally disordered speckle patterns[43–46].

The instabilities of the VAV lattices severely limit their potential applications. Nonlinearity may help to stabilize them[33,38], but it generally requires very high power densities, leading to unpredictable challenges in applications. Moreover, the nonlinearity mainly helps to balance diffraction or dispersion of the waves[47,48], rather than inhibiting instability-induced transverse jitter of the vortices. As a result, crystal-like VAV lattices have been created, thus far, in few optical nonlinear systems[33,38,49], remaining a challenge for the further work. Until now, stable VAV crystalline structures have not been reported in the linear propagation regime.

In this work, we investigate in detail nonlocal orbit-orbit couplings between the oppositely charged vortices embedded in a propagating coherent light field, and demonstrate, both theoretically and experimentally, an intriguing wave phenomenon of the VAV crystallization, supported by the balanced orbit-orbit coupling in the multi-VAV setting. We present a theoretical model analyzing the orbit-orbit couplings in such systems. The model includes a decoupling term, which accounts for the independent transverse motion of individual vortices, and a term, which nonlocally couples different vortices, with strength determined by the VAV spacing and propagation distance. The VAV crystallization stems from the balance of competing terms, as manifested by a flat phase distribution between adjacent VAV pairs. The VAV interactions considered here originate solely from the OAM-OAM (orbit-orbit) coupling in the light field, which takes place in the linear paraxial-propagation regime and does not require any light-matter interaction. The orbit-orbit coupling for stable lattices is basically distinct from nonlinear light-matter interactions, cf. refs. 33,49. It is also different from the extensively studied spin-orbit coupling, which refers to the interaction between the photonic spin and OAM[50–61]. While the spin-orbit coupling has been used in a broad range of applications[62–65], the nonlocal orbit-orbit coupling remains almost unexplored. Therefore, our approach and results open possibilities for manipulations with optical vortices and antivortices by engineering appropriate orbit-orbit couplings between them. In particular, the method for the creation of the robust phase-locked VAV structures in free space, reported in this work, can be utilized to enhance the capacity of optical communication and data-processing systems, and to manipulate multi-plane particle clusters, cf. ref. 66.

## Results

### The model and analytical solution

We start the presentation of the model, which produces phase-locked VAV lattices by appropriately arranging the initial geometric structures and engineering the orbit-orbit couplings. Specifically, we start by constructing the initial configuration of the complex light field composed of the multiple interactive vortices and antivortices. Mathematically, they can be realized by shaping an ambient field $G$ with two complex-valued polynomial functions[67,68]. The basic configuration is one with the opposite-charge vortices embedded in the Gaussian background $E(x,y)$ of width $w$. In this scenario, the initial VAV structure is given by

$$G = p(u) \cdot q(u^*) \cdot E \qquad (1)$$

where $u = x + iy \equiv re^{i\phi}$ defines the Cartesian and polar $(r, \phi)$ coordinates, and * stands for the complex conjugate. Complex polynomials,

$p(u) = \sum_{n=0}^{N} a_n u^n$ and $q(u^*) = \sum_{m=0}^{M} b_m u^{*n}$, represent a cluster composed of $N$ vortices and $M$ antivortices entangled with the background Gaussian field. The resultant VAV configuration is determined by the set of complex coefficients $a_n$ and $b_m$, which determine complex-valued roots $c_n$ and $d_m^*$ of polynomials $p$ and $q$, respectively. In turn, the roots define the position of each vortex pivot. Such initial configurations, built as VAV sets, usually cannot preserve their arrangement in the course of the propagation, due to the interplay between the opposite OAMs.

To reveal the orbit-orbit couplings, we investigate the propagation of the VAV configuration along the $z$ coordinate, according to the paraxial Schrödinger wave equation, $i\partial_z G = -(\lambda/4\pi)(\partial_x^2 + \partial_y^2)G$, with carrier wavelength $\lambda$. The evolution initiated by the input configuration (1) can be cast in an analytical form of[39]

$$G(x,y,z) = E(x,y,z)\left[F_0(x,y,z) + F_c(x,y,z)\right] \qquad (2)$$

where $E(x,y,z) = w^2|B(z)|/2 \exp(-B(z)r^2/2)$ represents the evolution of the Gaussian background, with

$$B(z) = 2\pi / \left[\lambda(z_R + iz)\right] \qquad (3)$$

and $z_R = \pi w^2/\lambda$ being the Rayleigh length. Further, term $F_0(x,y,z) = \prod_{n=1}^{N}\left[AB(z)u - c_n\right]\prod_{m=1}^{M}\left[AB(z)u^* - d_m^*\right]$, with $A \equiv w^2/2$, indicates that, due to the presence of $B(z)$, the vortices and antivortices move linearly in the transverse plane and stay separated in the course of the propagation. However, this term does not couple vortices and antivortices at different locations. The VAV orbit-orbit coupling is introduced by term $F_c = \sum_{k=1}^{N}(2A^2B(z) - 2A)^k k! P_{N,k} Q_{M,k}$, where $P_{N,k}(AB(z)u)$ and $Q_{N,k}(AB(z)u^*)$ are two $z$-dependent polynomial functions of variables $AB(z)u$ and $AB(z)u^*$. Expressions for the $P_{N,k}(ABu)$ and $Q_{N,k}(ABu^*)$ polynomials are displayed below in the Methods section, see Eqs. (6) and (7). On the contrary to $F_0$, the mixing term $F_c$ represents the interplay between the vortices and antivortices, which, in the framework of the linear propagation, leads to the mutual attraction, annihilation and repulsion between the vortices and antivortices, as well as the OAM Hall effect[39]. Note that the orbit-orbit coupling, emerging in the freely propagating paraxial light field, does not need the presence of any optical material, which makes this effect completely different from other photonic interactions, such as the above-mentioned spin-orbit coupling. The orbit-orbit coupling is effectively nonlocal, not constrained to the nearest-neighboring VAV pairs. It involves widely separated pairs too, with the long-range coupling strength gradually decaying with the increase of the separation[39,40]. Note that a recent work has introduced a parallel framework for separately describing the tilt, velocity and trajectory of each individual vortex, which can be applied to the cumbersome coupling of two oppositely charged vortices[36] and is compatible with our theory. As $F_c$ strongly depends on the propagation distance $z$, the orbit-orbit coupling is propagation-varying. As a consequence, the propagation dynamics of the VAV configurations are sensitive to the initial configurations.

In the following, we demonstrate the construction of equilibrium VAV configurations, by engineering the nonlocal orbit-orbit couplings. The spatial dependences of the mixing term $F_c$ allow us to appropriately arrange the configurations, and thus to design appropriate orbit-orbit couplings for producing robust crystalline VAV lattice. As demonstrated in the model of the vortex dipole, the coupling equilibrium of pivots in the course of the propagation can be achieved by adjusting the separation between them[36,39]. To this end, the lattice structure can be designed as an array composed of VAV pairs. We aim to arrange multiple vortex dipoles with co-orthogonal inclinations (horizontal and vertical) on a 2D grid with spacing $L$, the starting point being the central position, $(x,y) = (0, 0)$. The left panel in Fig. 1a illustrates the so constructed VAV crystalline patterns, which resemble the

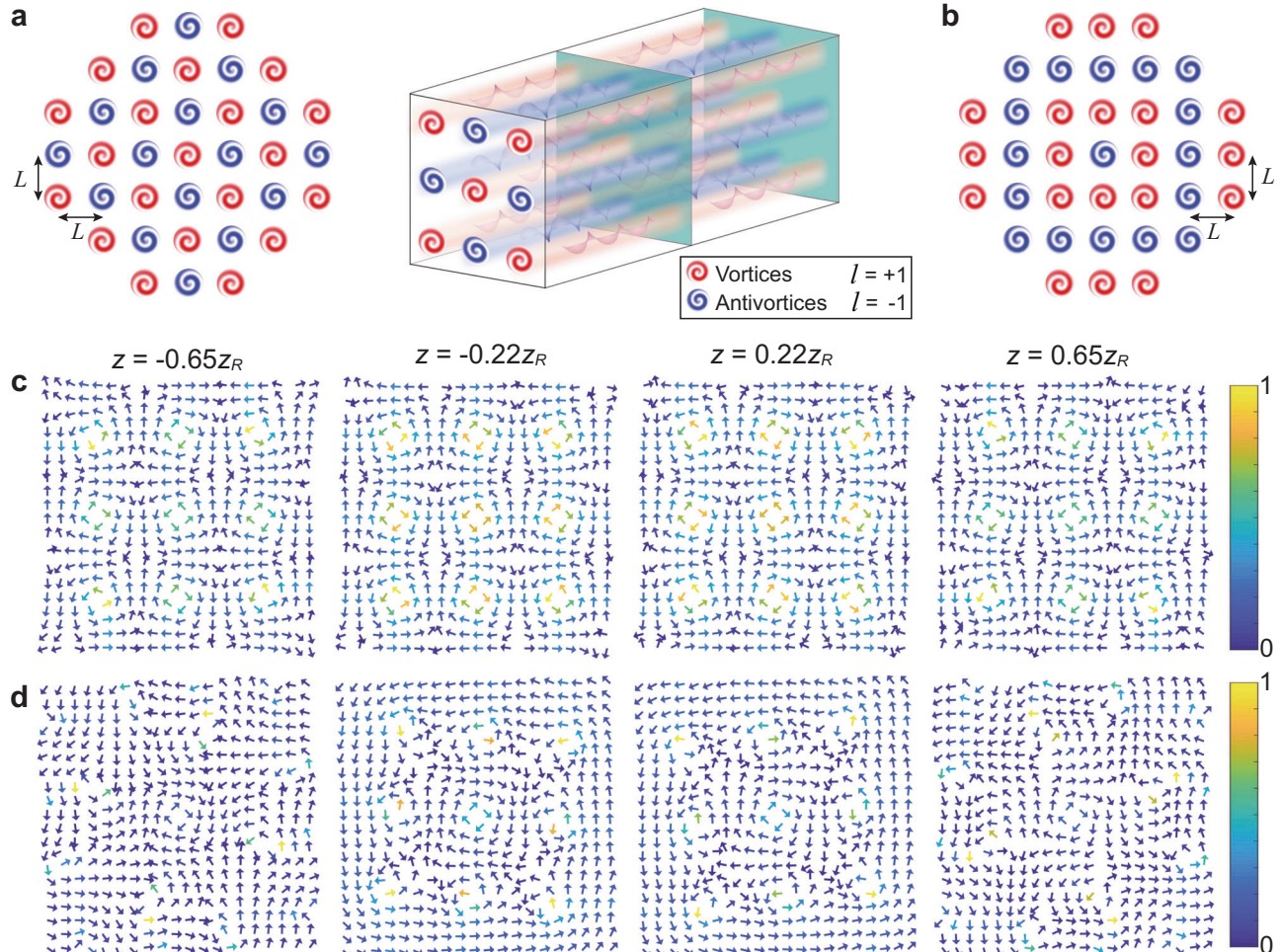

**Fig. 1 | Prediction of stable and unstable VAV (vortex-antivortex) lattice configurations. a** The left panel is the designed lattice structure (the Gaussian host field is not shown here), with alternating vortices and antivortices placed on the 2D grid. Red and blue elements designate the optical vortices and antivortices. The right panel illustrates the robust propagation of the lattice in the free space. **b** An unstable lattice configuration, which does not feature uniform alternation of the vortices and antivortices. **c** and **d** The phase structures of the VAV lattices introduced in panels **a** and **b**, respectively, produced by the analytical solution (Eq. (2)) with $w = 250\,\mu m$ and $L = 1.24w$. The panels **c** and **d** depict the evolution of the lattice phase patterns at different propagation distances: $z = −0.65z_R, −0.22z_R, 0.22z_R$ and $0.65z_R$, where $z_R$ is the Rayleigh length.

recently proposed 2D ionic square-shaped lattices, such as EuS[69], with the alternating vortices and antivortices playing the roles of positive and negative ions. Coordinates of the lattice sites, i.e., positions of the vortex and antivortex pivots, are given by roots $(c_n, d_m)$ of polynomials $P$ and $Q$. The right panel in Fig. 1a illustrates the robust propagation of the resultant 2D lattice configuration along the $z$ coordinate. We confirm this conclusion by showing in Fig. 1c the evolution of the phase structure of the complete field $G$ (see Eq. (1)) in the fragment of size $3 \times 3$ of the VAV lattice introduced in Fig. 1a. The fragment includes five vortices and four antivortices. The vortex polarity of each lattice element can be identified by the phase-gradient field, displayed by patterns of arrows in Fig. 1c. The phase varies rapidly close to the pivots, corresponding to phase singularities with the polarity of each one identified by the rotation of the gradient arrows. On the other hand, the phase becomes flat at boundaries between adjacent VAV pairs. The presence of the flat phase distributions is a manifestation of the VAV crystallization, as confirmed by the propagation-invariant phase-locked lattice structure.

Thus, the VAV crystallization is maintained by the local VAV alternation in the lattice. By contrast, other initial lattice structures lead to imbalanced orbit-orbit couplings, resulting in strongly unstable propagation dynamics. An example of an unstable lattice is provided by the square lattice, built as a central antivortex surrounded by

alternating vortex and antivortex layers, as shown in Fig. 1b. Although the lattice spacing is maintained in the course of the propagation of this input, we observe in Fig. 1d that the propagating phase pattern is strongly disturbed, featuring, in particular, annihilation and creation of VAV pairs.

## Experimental demonstration

Following the theoretical analysis, we have demonstrated the non-local orbit-orbit couplings between the vortices and antivortices and their crystallization in the experiment. In these contexts, a key point is to produce the interactive vortices and antivortices nested in the Gaussian envelope, using a computer-generated phase-only hologram. Experimental observation of the orbit-orbit couplings apparently was not clearly demonstrated in previous works, which used conventional techniques for generating multiple vortices and antivortices[41,70,71]. This objective is a challenging one as it requires to encode both the amplitude and phase information, which represents the coupling term $F_c$ in Eq. (2) for the VAV lattice in the Fourier space. At the initial position $z = 0$, the Fourier transform of the whole field is written as

$$\tilde{G}(k_x, k_y) = \tilde{E}_0(k_x, k_y) \cdot \left[ \tilde{U}_0(k_x, k_y) + \tilde{U}_c(k_x, k_y) \right] \quad (4)$$

where $\tilde{E}_0(k_x, k_y)$ is the Fourier transform of $E(x, y)$ at $z = 0$, with $k_x$ and $k_y$ being the corresponding spatial frequencies in the $(x, y)$ plane. Equation (4) includes two important terms, exhibiting similar mathematical form to the equation (2). However, these terms $\tilde{U}_0$ and $\tilde{U}_c$ are not the Fourier transforms of $F_0$ and $F_c$. They can be derived analytically as: $\tilde{U}_0 = \prod_{n=1}^{N}[iA(k_x + ik_y) - c_n]\prod_{m=1}^{M}[iA(k_x - ik_y) - d_m^*]$, and $\tilde{U}_c = \sum_{k=1}^{N}(-2A)^k k! \tilde{P}_{N,k} \tilde{Q}_{M,k}$, respectively. More details about $\tilde{U}_0$ and $\tilde{U}_c$ are presented below in Methods section. Very similarly, $\tilde{U}_0$ denotes the spatially decoupled vortices and antivortices in the Fourier space, while the $\tilde{U}_c$ couples them nonlocally, leading to the interactive elements. Thus, the correct Fourier transform of the interactive VAV lattice should comprise the non-coupling and coupling terms simultaneously. If the coupling term $\tilde{U}_c$ is not included in the phase mask, the produced vortices and antivortices would propagate independently without orbit-orbit coupling among them[41]. This important coupling term is essential and allows us to perform experiments for observing phenomena caused by the orbit-orbit couplings.

Based on the theory, we experimentally realized the above predictions by using the setup presented in Fig. 2a. A linearly polarized He-Ne laser beam with wavelength $\lambda = 632.8$ nm is appropriately expanded and collimated by using a beam expander (BE). The first beam splitter (BS) divides the laser beam in two: a reference beam and the other one, patterned by the phase spatial light modulator (Holoeye LETO II SLM, $1920 \times 1080$). The phase hologram (supplementary Sec. B) creating the interactive vortices and antivortices is realized by using the coding technique proposed by Bolduc[72], as specified in Methods section. Other efficient encoding techniques, such as the binary computer-generated methods[73,74] may also be utilized to produce the desired lattice patterns. In the experiment, we used a sufficiently broad Gaussian, to embed multiple vortices and antivortices. A typical phase hologram that encodes a $3 \times 3$ square lattice, comprising five vortices and four antivortices, which is shown in Fig. 2b, was uploaded into the SLM. The spatially modulated light beam, reflected from the mirror, passes through a focusing lens (with the focal length 500 mm) which performs the Fourier transform. The first-order diffractive beam of the hologram is selected by using an iris diaphragm, other diffractive beams being blocked. The generated VAV lattices and their interference patterns with another divided beam are then imaged by a charge-coupled device (CCD) mounted on an electrically controlled stage movable along the $z$ axis.

Figure 2c presents the preliminary experimental result, which amounts to the $3 \times 3$ square lattice, composed of nine elements, with the initial width $w = 250\,\mu m$ of the Gaussian holding beam, and lattice spacing $L = 1.28w$. Our experimental measurements show that the constructed VAV lattice can maintain its geometrical shape unchanged during evolution along a distance that is approximately three Rayleigh ranges (Fig. 2c displays the intensity patterns of the square lattice at four typical propagation distances). Polarities of individual vortices are identified through the measured phase distribution of the generated VAV-lattice in different propagation planes (Fig. 2d). The experimental method for the phase reconstruction is introduced in Methods section. More details for reconstructing the experimental phase of the $3 \times 3$ VAV lattice are given in supplementary section A. The measured intensity and phase distributions confirm the generation and propagation of the expected VAV lattice configuration in Fig. 1. Although positions of individual vortices slightly vary, the overall configuration keeps its shape in the course of the propagation, suggesting that the VAV lattice realizes a stationary pattern. We compare the experimental results with the theoretical predictions (Fig. 2e, f). Excellent agreements are observed, indicating the effect of the balanced orbit-orbit couplings which connect the lattice elements. We further find that the balance

of the couplings strongly depends on the lattice period, $L$. Indeed, varying $L$ may lead to disbalance between the orbit-orbit couplings, due to their nonlocality. For instance, settings of $L = 0.8w$ or $L = 2w$, the disbalanced couplings lead to the annihilation of vortices and antivortices, or separation between them, as shown in the supplementary section C. The simulations reveal the crystallization and robust propagation of the resulting VAV lattice in the interval of $1.1w < L < 1.3w$. In contrast with that, a lattice structure initially composed of nine vortices with identical polarities (in this case $F_c = 0$ in Eq. (2), indicating the absence of orbit-orbit couplings) starts to rotate and quickly disintegrates at an early stage of the propagation, see the corresponding result in the supplementary section D.

Next, we present examples of bigger VAV lattices which also demonstrate robust crystallization. One example is based on a $5 \times 5$ square lattice, in which elements at the corners are removed. The resulting robust lattice is composed of 9 vortices and 12 antivortices, with lattice spacing $L = 1.24w$, see Fig. 3a–d. The phase-only hologram used for the creation of this lattice is presented in the supplementary section B. The experimentally measured phase distributions of the produced lattices confirm the initial shape of the lattice, while Fig. 3a–d illustrate the self-maintained crystallized shape at different propagation distances, and the robust propagating phases are exhibited in Fig. 3e–h. However, the crystalline pattern is not completely stable, showing a weak trend toward fusion, starting from the edges of the lattice. Indeed, elements near the edges are slowly escaping, initiating an eventual transition from the crystalline state towards a turbulent one. This phenomenon is similar to the melting transition in solid-state crystals[69]. An example of the crystallization of a still bigger VAV lattice, of size $7 \times 7$, is displayed in Fig. 4a–d, and the corresponding phases are shown in Fig. 4e–h. Its VAV pattern is constructed on the grid from which three pivots are removed at the corners. The resulting lattice includes 21 vortices and 16 antivortices, with spacing $L = 1.24w$. This lattice structure is shown by the respective phase distribution in the supplementary section B. Additional measurements shown in Supplementary Sec. E directly demonstrate the crystallization process, which transforms a disordered VAV lattice into a regular one. At a late stage of the evolution, the melting of the VAV crystal starts from its edges, where individual elements tend to escape due to the gradual breakup of the balance of the orbit-orbit couplings, while the integrity of the core of the lattice is still maintained by the balanced competition between the couplings. Actually, the slow melting is caused by the gradual diffraction-driven expansion of the Gaussian background, as manifested by the $z$-dependent coefficient $B(z)$ in Eq. (3). These observations corroborate the theoretical prediction, as shown in Fig. 3i–p and Fig. 4i–p for the $5 \times 5$ and $7 \times 7$ lattices, respectively. The creation of still larger stable lattices is more challenging, due to the limited width of the Gaussian background.

To visualize the crystallizations and stable evolution of the VAV lattices considered above, experimentally recorded trajectories of all pivots in the lattices are presented in Fig. 5a–c. Nearly straight-line trajectories are clearly observed for the three robust VAV lattices. Accurate measurements show that individual propagation trajectories remain straight over a propagation distance up to $2.6z_R$. Figure 5d–f display projections of the 3D trajectories onto the transverse plane. These panels (in particular Fig. 5d–f) clearly indicate that vortex and antivortex pivots at the edge make the overall lattice disordered in the beginning; after that, the outer pivots gradually move onto the designated 2D grid, and then crystallize into a regular lattice structure which persists for a long propagation distance. Moreover, it is also confirmed that in the course of the disintegration, the pivots located closer to the core of lattice perform much slower motion than those residing at the edges of the lattice.

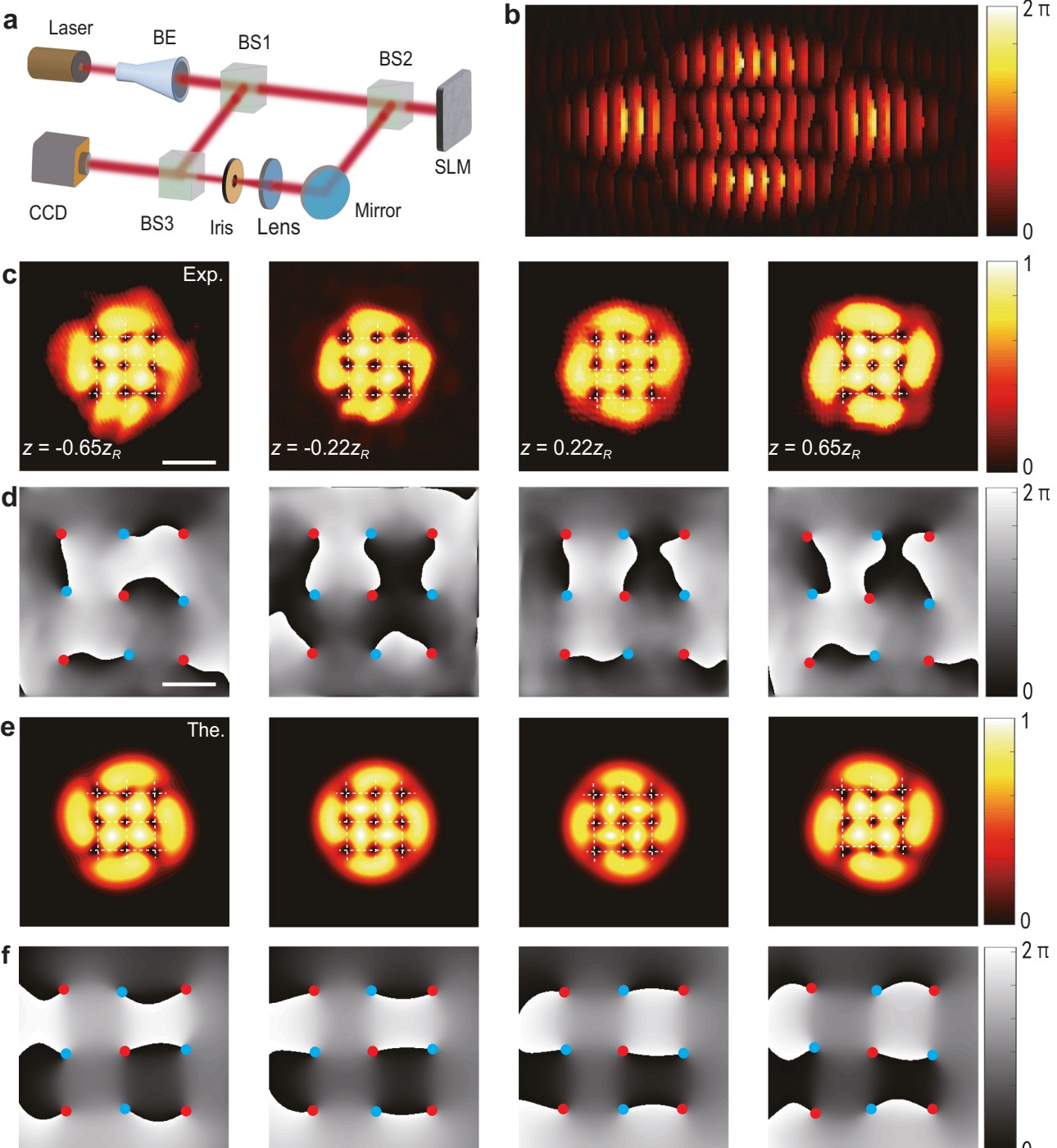

**Fig. 2 | Experimental observation of the crystallization of the VAV square lattice of the size in a 3 × 3. a** The experimental setup. A linearly polarized He-Ne laser with wavelength $\lambda$ = 632.8 nm is used. BE: beam expander, BS: beam splitter; SLM: spatial light modulator; CCD: charge-coupled device. **b** The computer-generated phase-only hologram used for the generation of the 3 × 3 VAV lattice. **c** Experimentally observed lattice patterns at different propagation distances: $z = -0.65 z_R$, $-0.22 z_R$, $0.22 z_R$ and $0.65 z_R$, with the width of the holding Gaussian beam $w$ = 250 $\mu$m and lattice spacing $L$ = 1.28$w$. **d** Measured phases corresponding to propagating fields at **c**. **e** and **f** Theoretical results for the intensity and phase distributions at the same propagation distances as in **c** and **d**, respectively. The white dashed lines in panels **c, e** depict the 2D grid, with the intersection points representing positions of pivots of the vortices and antivortices; the red and blue solid circles in panels **d, f** represent the vortices and antivortices, respectively. Panels **c, e** and **d, f** share the same scales, with scale bars being 0.60 mm and 0.28 mm, respectively.

We stress that solely the intensity and phase distributions are not sufficient to quantify the crystallization and stable propagation. We therefore have performed a detailed quantitative analysis on the stable propagation of the lattices by using the Pearson correlation coefficient (PCC)[75]. This makes it possible to quantify the VAV crystallization and identify the balanced nonlocal orbit-orbit couplings. PCC is defined as

a correlation between the intensity patterns $I_0(x, y)$ and $I_z(x, y)$ (here $I \equiv |G|^2$), recorded at $z = 0$ and at the current propagation distance:

$$\mathrm{PCC}(z) = \frac{\int\int (I_0 - \bar{I}_0) \times (I_z - \bar{I}_z) dx dy}{\sqrt{\int\int (I_0 - \bar{I}_0)^2 dx dy} \times \sqrt{\int\int (I_z - \bar{I}_z)^2 dx dy}} \quad (5)$$

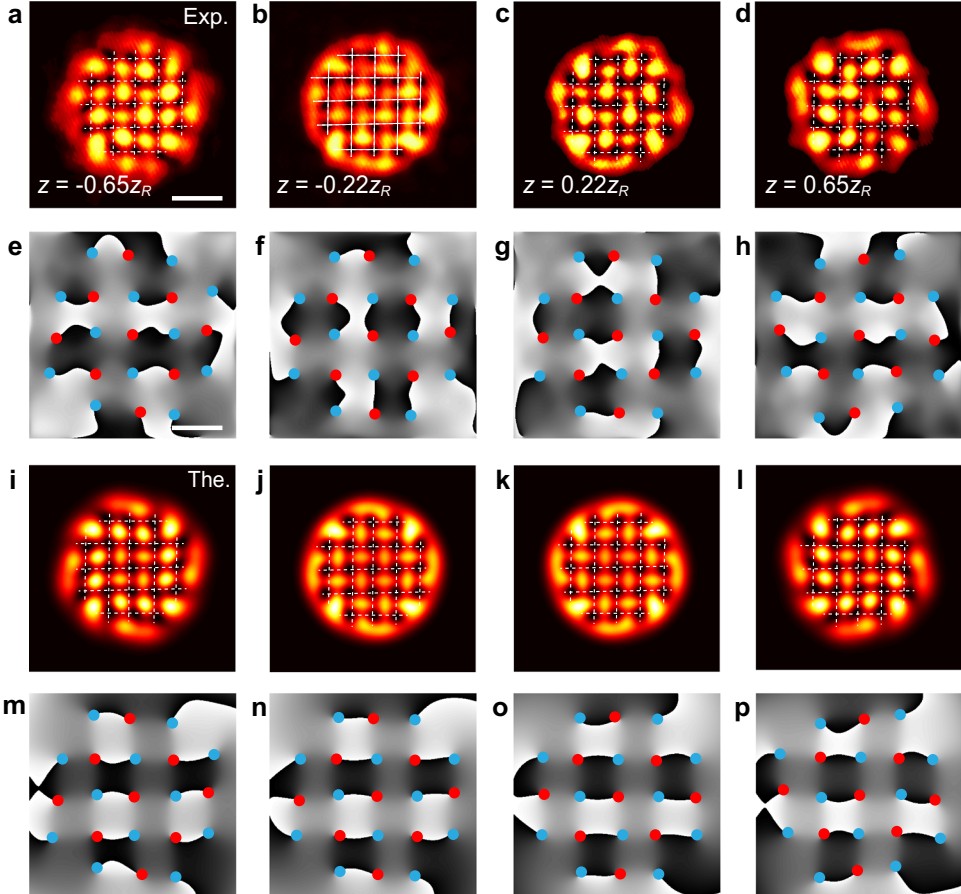

**Fig. 3 | The observation of the VAV crystallization of 5 × 5 square lattices.**
**a**–**d** Experimentally observed intensity distributions recorded at different propagation distances: $z = -0.65z_R$, $-0.22z_R$, $0.22z_R$, and $0.65z_R$. **e**–**h** The experimental phase distributions corresponding to **a**–**d**. The lattice spacing is set as $L = 1.24w$, with $w = 250\ \mu m$. These observations corroborate the corresponding theoretical predictions, shown in panels **i**-**l** (intensities) and **m**–**p** (phases), respectively. The intensity and phase panels share the same scale bars, being 0.70 mm and 0.42 mm, respectively.

where $\bar{I}$ is the average value of $I(x, y)$. The PCC coefficient takes values in the range between 0 and 1, larger ones indicating higher correlation between the two patterns. We adopt PCC = 0.8 as the critical value, so that PCC falling below it implies disintegration of the VAV lattice. Accordingly, we measure the PCCs of the robust lattices as a function of the propagation distance, as shown in Fig. 5g. It is seen that in all these cases the PCC keeps its value near 0.9, in agreement with the observation of the propagation-invariant lattice patterns. In particular, we note that the PCC initially gradually increases, reaching its maximum when the propagation distance changes from $z = -1.1z_R$ to $z = 0$. The slow increase of the PCCs indicates the formation of the VAV lattice. Afterwards, PCC is slowly decreasing, which implies that the crystal starts to melt. Thus, the observation of the nearly constant PCC suggests that the balanced orbit-orbit couplings maintain the robust lattice structures in the free space.

Finally, we demonstrate a counter-example of an unstable VAV lattice, to illustrate the imbalanced orbit-orbit coupling. This is a lattice composed of 7 × 7 pivots, shown in Fig. 1b, which does not feature the uniform alternation of vortices and antivortices in the horizontal and vertical directions, and has the same spacing as in Fig. 4a. Figure 6a shows the experimentally recorded intensity distributions of the lattice at $z = 0$, showing a regular VAV pattern which is essentially the same as in Fig. 4a. In drastic difference with the stable lattice, the present wrong one demonstrates no robustness in the course of the propagation. Under the action of imbalanced orbit-orbit couplings between the vortices and antivortices, the lattice undergoes dramatic structural changes in the course of the propagation.

Figure 6b, c make it obvious that the pattern quickly transforms into an irregular one, which may be considered as a turbulent optical state. Similar outcomes for the same propagation distances are produced by the theoretical solution in Fig. 6d–f. In Fig. 6g, the PCC value for the wrong structure demonstrates fast decay in the course of the propagation. It shows that the lattice structure survives only at a small distance, $z = 0.33z_R$, at which the PCC falls to the threshold value 0.8, which is defined above. Thus, the alternating VAV structure guarantees the balance of the orbit-orbit couplings for each vortex pivot, leading to long distances of the robust propagation, in contrast with the wrong lattices.

## Discussion
We have demonstrated that multi-VAV (vortex-antivortex) sets, embedded in the Gaussian host field, can crystallize into robust square-shaped lattices, which resemble ionic lattices in the solid-state physics. We have shown that the so constructed lattices preserve their structure in the free-space propagation over a distance essentially exceeding the Rayleigh (diffraction) length. We have presented the analytical model describing the vortex-antivortex crystallization, which results from the globally balanced orbit-orbit coupling acting upon each vortex or antivortex pivot. Eventually, due to the diffraction of the host field, such VAV crystals suffer gradual melting through the escape of individual elements from the edges of the lattice, while the core survives much longer propagation. Unlike the square-shaped VAV lattices, differently built ones suffer quick degradation, due to the action of imbalanced orbit-orbit couplings. It is plausible that the square-shaped lattices may be

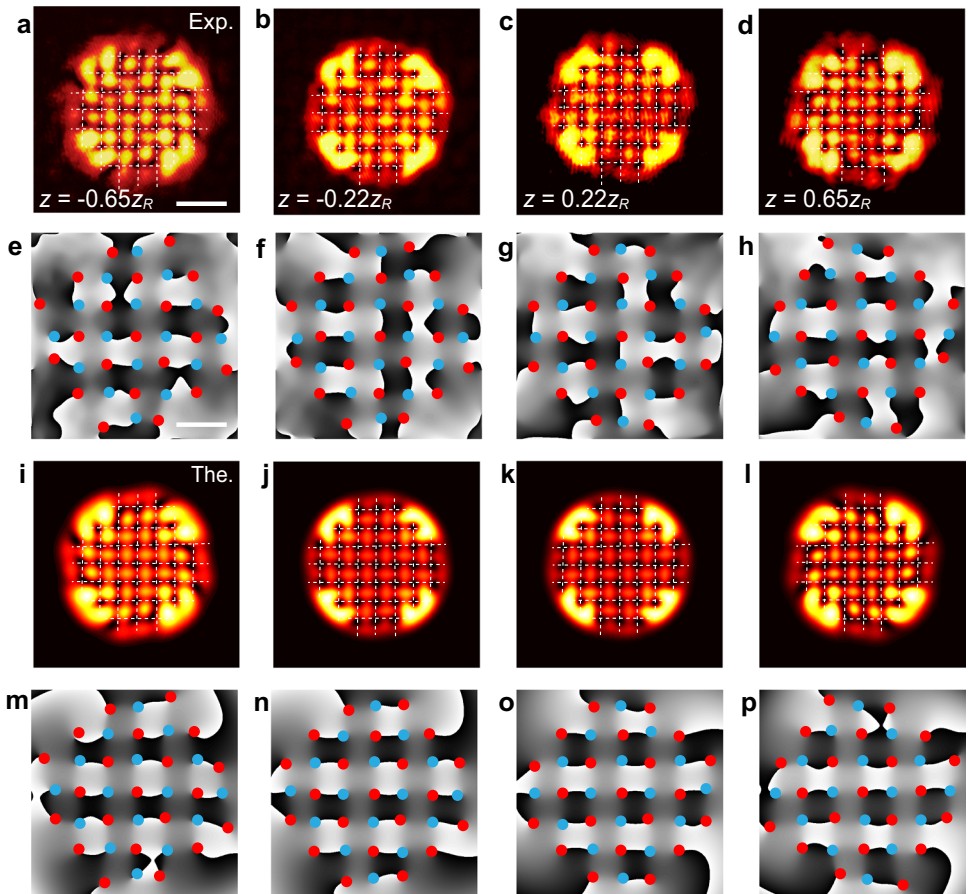

**Fig. 4 | The verification of crystallizing phenomenon of the 7 × 7 square lattices.** **a**–**d** The experimental intensity patterns detected at different propagation distance: $z = -0.65z_R$, $-0.22z_R$, $0.22z_R$, and $0.65z_R$. **e**–**h** The phase measurements corresponding to **a**–**d**. This lattice possesses the same spacing and Gaussian width as the case in Fig. 3. The measured results well match the theoretical predictions including the intensity and phase distributions shown in **i**–**l** and **m**–**p**, respectively. The intensity and phase panels share the same scale bars, being 0.86 mm and 0.55 mm, respectively.

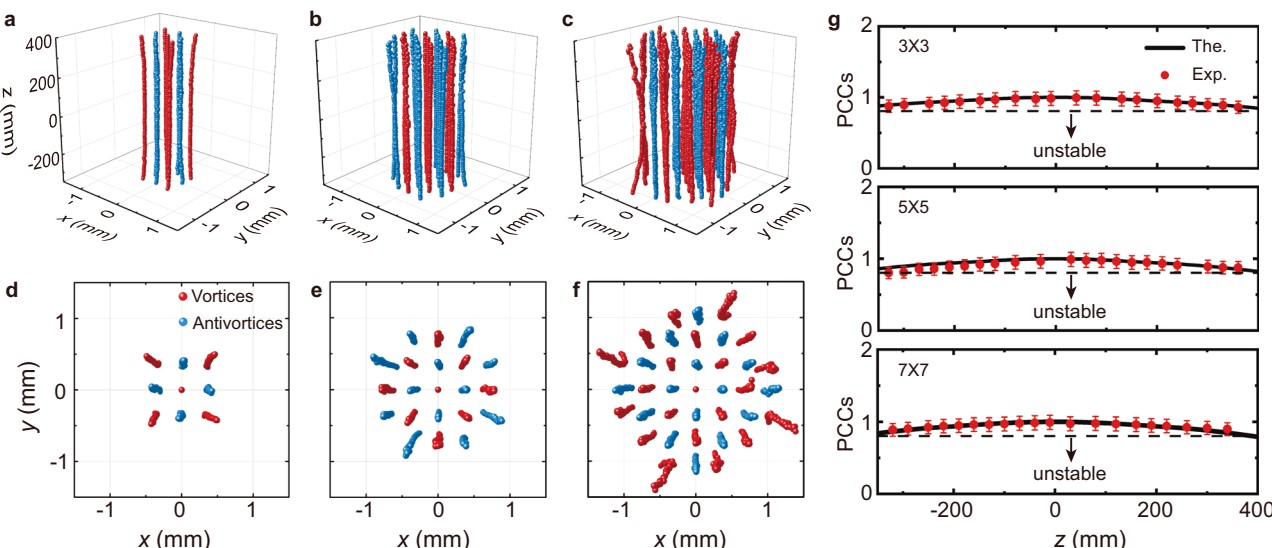

**Fig. 5 | Propagation trajectories of individual elements of the robust lattices, and the corresponding PCC as functions of z.** **a**–**c** The experimentally observed trajectories of motion of individual vortices and antivortices in the three above-mentioned square-shaped VAV lattices: **a** 3 × 3, **b** 5 × 5, and **c** 7 × 7. **d**–**f** Projections of the trajectories from panels **a**–**c** onto the transverse (x, y) plane. In **a**–**f**, the red data represents results for the vortices while the blue data for the antivortices. **g** The measured (data points) and theoretically predicted (solid curves) values of PCC for the different lattices. The dashed lines in **g** indicate the critical point at which PCC = 0.8. Below it, the pattern becomes turbulent and unstable. Values of experimental data in **g** are means plus s.e.m (10%).

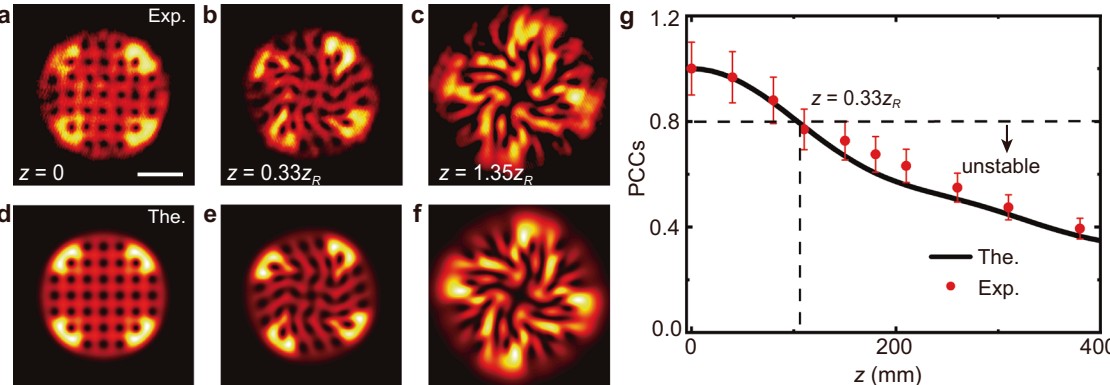

**Fig. 6 | Observation of imbalanced nonlocal orbit-orbit couplings.** The imbalanced interplay between the vortices and antivortices leads to transition of the regular pattern into a turbulent one. **a–c** The experimentally measured intensity distributions of the unstable $7 \times 7$ lattice, whose initial configuration is displayed in Fig. 1b. Experimental parameters $L$ and $w$ are the same as in Fig. 4. **d–f** Theoretical results corresponding to the experimental ones in panels **a–c**. **g** The experimental (red data point) and theoretical (black curve) PCC values of the lattice as a function of $z$. Values of experimental data in **g** are means plus s.e.m (10%). Panels **a–f** share the same scale, with scale bar shown in **a** being 1 mm.

further stabilized against melting by the inclusion of moderately strong self-focusing nonlinearity. Such a nearly stable lattice configuration, used as an input, can significantly reduce the light intensity required for the formation of fully stable nonlinear optical crystals[76]. Due to the universal nature of the paraxial wave propagation, robust configurations based on the balanced orbit-orbit couplings and the resulting VAV crystallization may be expected in other physical systems, such as matter waves[77], electron beams[78], acoustics[79] and hydrodynamics[80].

It is relevant to stress once again that, while manipulating light fields by means of spin-orbit couplings has drawn much interest[56], the orbit-orbit couplings acting between vortices and antivortices, reported here, remained unnoticed. Thus, our results offer a useful coupling scheme for manipulations with vortices and antivortices. In particular, we have presented a reliable optical emulation of 2D ionic-like crystals (note that there are very few real solid-state settings, which admit the existence of 2D square-shaped ionic lattices[69]). The creation of more sophisticated stable vortex-antivortex lattices can be expected by means of appropriate orbit-orbit couplings, in addition to the fluidity demonstrated in Refs. 81,82. In this vein, the concept of effective phase diagrams can be put forward for describing phase transitions in the structured light[33]. The phase diagrams of the VAV structures can therefore emulate different condensed-matter phases – in particular, for identifying the general crystallization process (disorder-to-order transitions). Furthermore, kinetics mediated by lattice defects (for instance, in graphene[83]) can be plausibly also emulated in optical VAV lattices. In terms of potential applications, the stable VAV lattices are very promising media in optical communications and all-optical data processing, as the lattices make it directly possible to enlarge the channel capacity[6–8].

## Methods
### Expressions of the propagating polynomial functions
In this section, we present expressions of the propagating polynomial functions both in the real and Fourier spaces. We start by considering the propagation of the initial vortex-antivortex lattice represented by Eq. (1). A general solution to the paraxial Schrödinger equation can be expressed as follows

$$G(x,y,z) = \mathcal{IFT}\left\{\tilde{G}\left(k_x, k_y\right) \exp\left[-\frac{i}{2k_0}\left(k_x^2 + k_y^2\right)z\right]\right\}$$

where $k_0 = 2\pi/\lambda$ is the free-space wavenumber, and $\mathcal{IFT}\{\cdot\}$ denotes the inverse Fourier transform operator, and $\tilde{G}\left(k_x, k_y\right) = \mathcal{FT}\{G(x,y,z=0)\}$

is the Fourier transform of the input at $z = 0$. Based on the known property for the Fourier transform $\mathcal{FT}$,

$$\mathcal{FT}\left\{(x \pm iy)^n G(x,y)\right\} = \left[i\left(\frac{\partial}{\partial k_x} \pm i\frac{\partial}{\partial k_y}\right)\right]^n \tilde{G}\left(k_x, k_y\right)$$

the Fourier transform of the initial configuration can be written as

$$\tilde{G}\left(k_x, k_y\right) = \sum_{n=0}^{N} a_n (iD)^{N-n} \times \sum_{m=0}^{M} b_m \left(iD^*\right)^{M-m} \times \tilde{E}_0(k_x, k_y)$$

Here the complex differential operator is $\tilde{D} = \partial/\partial k_x + i\partial/\partial k_y$, and $\tilde{E}_0 = (w^2/2)\exp\left[-w^2\left(k_x^2 + k_y^2\right)/4\right]$ is the Fourier transform of the Gaussian background at the initial position. It implies that $\tilde{G}(k_x, k_y)$ is a superposition of many different components of light field in the Fourier space, written as $\tilde{G}(k_x, k_y) = \sum_{n=0}^{N} \tilde{G}_n(k_x, k_y)$. The summation of these Fourier series leads to an explicit form,

$$\tilde{G}\left(k_x, k_y\right) = \tilde{E}_0\left(k_x, k_y\right) \times \left[\tilde{U}_0\left(k_x, k_y\right) + \tilde{U}_c\left(k_x, k_y\right)\right]$$

where $\tilde{U}_0 = \prod_{n=1}^{N}\left[iA(k_x + ik_y) - c_n\right]\prod_{m=1}^{M}\left[iA(k_x - ik_y) - d_m^*\right]$, and $\tilde{U}_c = \sum_{k=1}^{N}(-2A)^k k! \tilde{P}_{N,k}\tilde{Q}_{M,k}$, respectively, with $\tilde{P}_{N,k}$ and $\tilde{Q}_{M,k}$ being

$$\tilde{P}_{N,k} = \sum_{l=0}^{N-k} a_l C_k^{N-l}\left[iA\left(k_x + ik_y\right)\right]^{N-k-l}$$

$$\tilde{Q}_{M,k} = \sum_{l=0}^{M-k} b_l C_k^{M-l}\left[iA\left(k_x - ik_y\right)\right]^{M-k-l}$$

Accordingly, the propagating light field is represented by the inverse Fourier transform of $\tilde{G}(k_x, k_y)$, to which the propagation operator is applied. It yields,

$$G(x,y,z) = E(x,y,z) \times \left[F_0(x,y,z) + F_c(x,y,z)\right]$$

where $E(x,y,z) = w^2|B|/2\exp\left(-Br^2/2\right)$ accounts for the Gaussian envelope evolution, with $B = 2\pi/[\lambda(z_R + iz)]$ and $z_R = \pi w^2/\lambda$ being the Rayleigh length, as defined above. Here $F_0 = \prod_{n=1}^{N}(ABu - c_n)\prod_{m=1}^{M}(ABu^* - d_m^*)$, with $A = w^2/2$, denotes the decoupling term, and $F_c$ represents the orbit-orbit couplings. It is written as $F_c = \sum_{k=1}^{N}\left(2A^2 B - 2A\right)^k k! P_{N,k} Q_{M,k}$, where $P_{N,k}(ABu)$ and $Q_{N,k}(ABu^*)$ are two propagation-dependent polynomial functions of arguments $(ABu)$

and $(ABu^*)$, expressed as

$$P_{N,k} = \sum_{l=0}^{N-k} a_l C_k^{N-l}(ABu)^{N-k-l} \qquad (6)$$

$$Q_{M,k} = \sum_{l=0}^{M-k} b_l C_k^{M-l}(ABu^*)^{M-k-l} \qquad (7)$$

It is interesting to find that the Fourier transform of the input field $\tilde{G}(k_x, k_y)$ exhibits similar form to the solution represented in the real space. The essential terms $\tilde{U}_0$ ($F_0$) and $\tilde{U}_c$ ($F_c$) represents decoupling and mutual coupling between the vortices and antivortices in the Fourier (real) space. However, we should note that $\tilde{U}_0$ and $\tilde{U}_c$ are not the direct Fourier transform of $F_0$ and $F_c$.

## The generation of the phase-only hologram

The computer-generated hologram encoding both the phase and amplitude information of the VAV lattice can be generated by means of the phase-only modulation technique. This requires to derive an analytical solution for the optical lattice in the Fourier domain. As mentioned above, at the initial position, the Fourier spectrum of the entire field is given by $\tilde{G}(k_x, k_y) = \tilde{E}_0(k_x, k_y)[\tilde{U}_0(k_x, k_y) + \tilde{U}_c(k_x, k_y)]$, which can be rewritten as

$$\tilde{G}(k_x, k_y) = \tilde{G}_0(k_x, k_y) \exp\left[i\tilde{\Phi}(k_x, k_y)\right]$$

where $\tilde{G}_0$ and $\tilde{\Phi}$ represent the amplitude and phase of $\tilde{G}$. Note that $\tilde{E}_0$ is a real function and phase $\tilde{\Phi}$ originates from the decoupling term $\tilde{U}_0$ and the coupling one $\tilde{U}_c$. The overall phase and amplitude of $\tilde{G}(k_x, k_y)$ are encoded into the phase-only hologram[72], as specified in the following formula:

$$H(k_x, k_y) = M(k_x, k_y) \times \text{Mod}\left[\varphi(k_x, k_y) + \frac{2\pi k_x}{\Lambda}, 2\pi\right]$$

where $M(k_x, k_y) = 1 + \text{sinc}^{-1}[\tilde{G}_0(k_x, k_y)]/\pi$, and $\varphi(k_x, k_y) = \tilde{\Phi}(k_x, k_y) - \pi M(k_x, k_y)$. Here $\text{Mod}(\cdot)$ denotes a modulo operation, and $\text{sinc}(x) = \sin(x)/x$. Note that the hologram includes a blazed grating, which is utilized to diffract the target light field onto the first-order component of the hologram. In the experiment, the periodicity of the grating in the $x$ direction is $\Lambda = 64\,\mu m$. One can use other phase-only modulation techniques to generate the optical holograms for the creation of the VAV lattices[73,74].

## Experimental details for the observation of the VAV crystallization

First, regarding the generation of the phase-only optical masks, we emphasize that the coupling term $F_c$ in Eq. (2) in the main text should be considered in the framework of the phase-only modulation technique. While the VAV lattice can be generated by implementing the phase-only hologram without encoding the term $\tilde{F}_c$, the generated vortices and antivortices would not interact via the nonlocal orbit-orbit couplings. Second, since the orbit-orbit coupling is very sensitive to the initial lattice configuration, we have built a recursion algorithm to address the inverse function of $\text{sinc}(\cdot)$ for the implementation of the phase-only modulation technique. This is important for generating a high-quality phase-only hologram. Otherwise, the obtained phase-only mask is less accurate for observing the VAV crystallization. Considering the sinc function with value ranging between 0 and 1, we normalize the amplitude expression $\tilde{G}_0$ to match the function $\text{sinc}^{-1}(\cdot)$. Finally, as the VAV lattice is nested in the Gaussian envelope, in the experiment, we had to choose an appropriate beam waist, to improve the quality of the interactive VAV lattice. Moreover, the SLM device requires an input plane wave, while the laser is working in its fundamental Gaussian mode. Therefore, the Gaussian envelope was expanded properly to cover the whole SLM screen.

## The procedures for the phase reconstruction

This method can recover the phase of the experimentally-generated field through the single shot of the interference between the objective and plane waves[84]. The measured intensity pattern produced by the superposition of the reference wave $R(x, y) = R_0 \exp(ik_c x)$ and object $G(x, y) = G_0(x, y) \exp[i\psi_G(x, y)]$ ($G_0$ and $\psi_G$ represent the amplitude and phase, respectively) can be expressed as

$$I(x, y) = |R_0|^2 + |G_0|^2 + R_0\left[G\exp(-ik_c x) + G^*\exp(ik_c x)\right]$$

where $k_c$ denotes the carrier frequency. We note that the third term representing the interference fringes is determined by the objective and the carrier-wave phases. To extract the phase distribution $\psi_G(x, y)$, the Fourier transform of the interference pattern is performed. Considering the fact that a phase variation in the real space causes a frequency displacement in the Fourier domain, we obtain

$$\mathcal{FT}[I(x, y)] = \tilde{I}_1\left(k_x, k_y\right) + R_0\left[\tilde{G}\left(k_x - k_c, k_y\right) + \tilde{G}^*\left(-k_x - k_c, -k_y\right)\right]$$

where $\tilde{I}_1\left(k_x, k_y\right) = \mathcal{FT}\left(|R_0|^2 + |G_0|^2\right)$. It it evident that the term with displacement of $k_c$ in the frequency domain affects the objective's Fourier transform, while its counterpart with the identical shift to the other side is a conjugate one. A square filter centering at $(k_c, 0)$ is applied to identify the Fourier distribution of the objective field. Then, the inverse Fourier transform is performed, recovering both the amplitude and phase. As a result, the imaginary part of the logarithm of the recovered field yields the measured phase. Note that the so produced filtered field is shifted to the origin point in the frequency domain, in order to recover the vortex located at the center. A relevant example is presented in the supplementary section A. Negligible experimental errors result from the diffracted phases of the reference wave and the Fourier lens, slightly distorting the objective phase.

## Data availability

All data that supports the plots within this paper and other findings of this study are available from the corresponding authors (S. F., Z. L. and Z. C.).

## Code availability

The custom code used in this study is available from the corresponding authors (S. F., Z. L., and Z. C.).

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

## Acknowledgements

We acknowledge support from the National Natural Science Foundation of China (nos. 12374306 (to S. F.), 62175091 (to Z. L.)), the Pearl River talent project (no. 2017GC010280 to S.F.), the Key-Area Research and Development Program of Guangdong Province (no. 2020B090922006 to Z.C.), the Guangzhou science and technology project (no. 202201020061 to S.F.), and the Israel Science Foundation (no. 1695/22 to B.M.).

## Author contributions

S.F. conceived the concept. S.F. and B.A.M. carried out the analytical considerations. S.F., H.L., Y.L., and B.A. Malomed drafted and revised the paper. H.L., Y.L. and J.R. performed the experiments. H. Lin and Y. Liao performed numerical simulations and designed the phase-only holograms. Z.C. supervised the project. All authors participated in discussions and contributed to the editing of the article. H.L. and Y.L. contributed equally to this work.

## Competing interests

The authors declare no competing interests.
