## [Peer Review File · Nature Communications]

Optical vortex-antivortex crystallization in free spaceREVIEWER COMMENTS

Reviewer #1 (Remarks to the Author):

This work is a combined theoretical and experimental analysis of stable arrays of optical vortices within a non-interacting photon regime. The experimental work (not my specialty) seems to be carried out cleanly, and the theoretical analysis is relatively easy to follow, mostly amounting to interpretation of the beam structure giving rise to arrays of singularities. The writing is lucid, the paper is of a reasonable length, and it is well laid out, a good read for those interested in the topic.

However, there is a significant problem, in my opinion, with the key assertions made by the authors that causes me to recommend that the paper not be published without being substantially revised. Perhaps the primary claim of this work is that “Until now, stable VAV crystalline structures have not been reported in the linear propagation regime.” There is a plethora of published work that makes it clear that this is simply not correct. A few key examples are provided below, but this is an area with a rich literature spanning over a decade.

Ma et al., Generation of Circular Optical Vortex Array, *Annalen der Physik (Berlin)* 529, 1700285 2017
<https://doi.org/10.1002/andp.201700285>

Li et al., Generation of optical vortex array along arbitrary curvilinear arrangement, *Optics Express* 26(8) 2018
<https://doi.org/10.1364/OE.26.009798>

Wang et al., Tailoring a complex perfect optical vortex array with multiple selective degrees of freedom, *Optics Express* 29(7) 10811 2021
<https://doi.org/10.1364/OE.422301>

Lusk et al., Quantized optical vortex-array eigenstates in a rotating frame, *Physical Review A* 108, 023509 2023
<https://doi.org/10.1103/PhysRevA.108.023509>

A second claim of the paper is that vortices of opposite charge uniformly annihilate or repel in linear media, but that is not the case either as is clear from the works cited above.

A third area that does not link with the state of the art is the explanation offered for how vortices of opposite charge interact. This is a substantial portion of the theoretical analysis offered. However, this has been explained, quite generally, in terms of the interaction of tilted vortices in Ref [19] of the manuscript (Andersen et al., *Physical Review A* 2021 <https://doi.org/10.1103/PhysRevA.104.033520>). That work is also relevant to the analysis offered in the manuscript as to how vortices interact to stabilize their positions within an array. The manuscript setting is an array version of the individual pairs considered in this earlier publication.

There is a lot to like and learn from both the theory and experimental work offered in this manuscript. However, the relevant body of literature on this topic is seemingly missed in favor of citing parallel developments in quantum fluids. A reasonable course of action is for the authors to

assiduously review the field of linear optical vortex arrays to identify novel aspects of their research that can be consolidated into a cornerstone for a much-revised manuscript and subsequently well-received addition to the field.

As an aside, I would think that the very nice apparatus developed by the authors could be used to investigate arrays as analogs to two-dimensional condensed matter. For instance, phase diagrams could be developed to identify stable and unstable crystals associated with different arrangements of two-atom structures. Phase transitions could then be studied as well as order-disorder transformations. Polycrystalline materials could be constructed to observe and quantify grain boundary dynamics, allowing one to quantify surface tension at grain boundary surfaces and the kinetics of grain boundary triple points. Defects could be introduced and their kinetics compared, for instance, with defect structures in graphene lattices (<https://doi.org/10.1103/PhysRevLett.100.175503>). Leaning into the direction of optical matter seems to offer more promise than the vague nod made to optical communications that is often invoked by those more interested in underlying physics than the technological applications.

Reviewer #2 (Remarks to the Author):

The authors claim to observe "Optical vortex-antivortex crystallization in free space" (free space meaning that it is not a nonlinear effect). If they really saw crystallization in free space, I would agree that this is a very interesting result – but they don't. They show that vortex-antivortex lattices are mostly stable and they call this "crystallization", but I think that "crystallization" implies a process in which a crystalline structure is formed. In contrast, the authors are simply showing that the structure is maintained.

It looks like the authors agree with my definition because in the abstract, they say "Here, we demonstrate that multiple optical VAV clusters nested in the propagating coherent field can crystallize into new patterns which preserve their lattice structure in the course of the linear propagation over distance up to several Rayleigh lengths." But in their data, they are looking for a maintained shape – I don't see a single figure that starts with a random or disordered configuration and observes a lattice appearing with propagation (i.e. "crystallization"). The only thing I see is Fig. 4g, which shows a very small increase in the "PCC" as the beam goes through a focus, but the increase is still within the error bars and the vortex lattice structure is not discernibly different (there seems to be more rotation in the vortices further from the waist, but the vortex centers don't seem to be significantly further apart). Furthermore, it is unclear from Fig 2a where the imaging plane of the SLM is; if it is at $z=0$ (as I suspect), then it is trivial that the best PCC is here, since any defects will cause distortions away from the imaging plane.

Other comments:

1. The second paragraph closes with "The instability spontaneously transforms initially regular lattices into disordered speckles [21–23]." I looked at these references, and it looks like these three papers do not deal with propagation dynamics of regular lattices. I don't see how any of them supports the claim that the authors make. In contrast, this paper shows that the only way to get disordered speckle from a lattice is to disorder the lattice (so that it isn't a lattice anymore)
A Balbuena Ortega, S Bucio-Pacheco, S Lopez-Huidobro, L Perez-Garcia, FJ Poveda-Cuevas, JA Seman,

AV Arzola, and K Volke-Sepúlveda. Creation of optical speckle by randomizing a vortex-lattice. *Optics express*, 27(4):4105– 4115, 2019.

2. At the end of the third paragraph, the authors state "Until now, stable VAV crystalline structures have not been reported in the linear propagation regime." In addition to the paper mentioned above, what about this paper?

Alexander Dreischuh, Sotir Chervenkov, Dragomir Neshev, Gerhard G Paulus, and Herbert Walther. Generation of lattice structures of optical vortices. *JOSA B*, 19(3):550–556, 2002.

3. Optical vortices often coincide with dark spots in a beam, but not always so. The authors should invest the time and equipment necessary to build an experimental means of measuring the phase, where the vortices can be directly measured and the VAV character can be observed. Surprisingly, Fig 3 shows multiple beamsplitters, which suggests that the authors are set up for interferometric phase measurements. I think that Nat Comm readers expect to see phase in current papers.

Editors, Nature Communications
Re: MS No. NCOMMS-23-55028-T
"Optical vortex-antivortex crystallization in free space"
by Haolin Lin *et al.*

We thank the referees for the detailed assessment of our manuscript. Below, we provide point-by-point responses to all comments of Referees, and summarize changes made in the revised manuscript. All the changes are highlighted by the red font in the revised text.

Reviewer #1 (Remarks to the Author):

This work is a combined theoretical and experimental analysis of stable arrays of optical vortices within a non-interacting photon regime. The experimental work (not my specialty) seems to be carried out cleanly, and the theoretical analysis is relatively easy to follow, mostly amounting to interpretation of the beam structure giving rise to arrays of singularities. The writing is lucid, the paper is of a reasonable length, and it is well laid out, a good read for those interested in the topic.

Reply: We appreciate the positive appraisal of the manuscript.

However, there is a significant problem, in my opinion, with the key assertions made by the authors that causes me to recommend that the paper not be published without being substantially revised. Perhaps the primary claim of this work is that "Until now, stable VAV crystalline structures have not been reported in the linear propagation regime." There is a plethora of published work that makes it clear that this is simply not correct. A few key examples are provided below, but this is an area with a rich literature spanning over a decade. Ma et al., Generation of Circular Optical Vortex Array, *Annalen der Physik (Berlin)* 529, 1700285 2017; Li et al., Generation of optical vortex array along arbitrary curvilinear arrangement, *Optics Express* 26(8) 2018; Wang et al., Tailoring a complex perfect optical vortex array with multiple selective degrees of freedom, *Optics Express* 29(7) 10811 2021; Lusk et al., Quantized optical vortex-array eigenstates in a rotating frame, *Physical Review A* 108, 023509 2023.

Reply: We thank the Reviewer for drawing attention to the relevant papers. Actually, they did not directly address square-shaped VAV lattices: the first work introduced the VAV lattices with concentric circular layers; the other mentioned papers introduced vortex arrays with structures which are totally different from those investigated in our manuscript. The mentioned works essentially focused on techniques used for the creation of vortex arrays, rather than their propagation phenomenology and coupling between vortex and antivortex singularities. A robust vortex array with rotational characteristic of a rigid body was recently demonstrated in *Phys. Rev. A* 108, 023509 (2023) (Ref. [19] in revised main text), but positions of off-axis vortex pivots were varying with the propagation distance. The stability of those lattices in the course of long-range propagation was not demonstrated cogently. For instance, in the theoretical paper, *Annalen der Physik* 529, 1700285 (2017) (mentioned by the Reviewer and added to the revised manuscript as Ref. [20]), a circular vortex array is based on two concentric optical vortices with different radii. The array is strongly distorted in the course of the free-space propagation, with the constituent vortex singularities moving

away from the center (please see Fig. S1 included here, which demonstrates the instability of this configuration, but would not be relevant in the revised paper). Besides that, the theory presented in section Material and Methods has clearly demonstrated an important fact that the formation and persistence of the VAV arrays is maintained, in our work, by the well-designed arrangement, *viz.*, the square lattice, whereas the circular lattice in Fig.S1 definitely does not persist.

We thank the Reviewer for pointing out these references, which are all cited in the revised manuscript (see Refs. [19-22]).

Fig.S1: The propagation of the circular vortex array composed of two perfect optical vortices. **a** The intensity patterns at different propagation distances: $z = 0, 20$ and 40 cm.

b The phase distributions corresponding to **a**. In the plane of $z = 0$, one high-order vortex is placed at the center, with topological charge of $l = +5$ (the red circle in **b1**), and other 10 concyclic antivortices (blue circles in **b1**) are arranged around the center point. These panels indicate the deformation of the circular vortex lattice in course of propagation. The simulated results are based on the theoretical model reported in Ref. [20] (the figure is not included in the main text, as it is not necessary there).

A second claim of the paper is that vortices of opposite charge uniformly annihilate or repel in linear media, but that is not the case either as is clear from the works cited above.

Reply: This relevant comment calls for an explanation. If it is just one vortex dipole (VAV pair) embedded in the Gaussian background, in the general case the free-space propagation process eventually results in VAV annihilation or repulsion, featuring nontrivial coupling between the vortex and antivortex. We have observed this phenomenon in recent work [39]. Furthermore, in the case of double vortex dipoles with different arrangements, restoration of VAV pairs, which may be understood as a manifestation of the intrinsic OAM Hall effect, was also demonstrated. The intrinsic coupling is determined by the distance between the vortex' and antivortex' pivots and spatial geometry of the VAV configuration, which suggests the possibility to realize the coupling balance, as demonstrated in this work. This point has been clarified in the revised manuscript, please see Page 2.

A third area that does not link with the state of the art is the explanation offered for how vortices of opposite charge interact. This is a substantial portion of the theoretical analysis offered. However, this has been explained, quite generally, in terms of the interaction of tilted vortices in Ref [19] of the manuscript (Andersen et al., Physical Review A 2021). That work is also relevant to the analysis offered in the manuscript as to how vortices interact to stabilize their positions within an array. The manuscript setting is an array version of the individual pairs considered in this earlier publication.

Reply: We thank the Review for highlighting the similarity to the paper cited in our original manuscript. The theory presented in that paper introduced an appropriate framework for separately describing the tilt, velocity and trajectory of individual vortices, which helps to understand the cumbersome coupling of two oppositely charged vortices. As concerns the similarity, the analytical solution for an arbitrary VAV array is derived in our manuscript, as $G(x, y, z) = E \times (F_0 + F_c)$ (i.e., $\psi = \psi_{bg}\psi_v$, which is similar to Eq. (3) in Ref. [19], which corresponds to Ref. [36] in the revised main text). Hence, our theoretical approach and the one elaborated in Ref. [19] are indeed congruent.

On the other hand, the difference between the theory presented in Ref. [36] and ours is evident. That work provided a local approach for the investigation of the coupling between the vortex' tilt and gradient in the Gaussian background field, starting from kinematics of individual pivots, and deriving a precise model after implementing some complicated procedures. In contrast, we find that $\psi_v = F_0 + F_c$, which indicates that the interaction between all vortices and antivortices is governed by the nonlocal coupling term F_c , hence the coupling is explained by the global analysis, which was not presented in Ref. [36]. Nevertheless, these distinct theoretical frameworks are compatible, as evident from the similar vortex-antivortex pair dynamics demonstrated in Ref. [36] and our previous work [39].

These points have been clarified in the revised manuscript (please see highlighted text on page 4).

There is a lot to like and learn from both the theory and experimental work offered in this manuscript. However, the relevant body of literature on this topic is seemingly missed in favor of citing parallel developments in quantum fluids. A reasonable course of action is for the authors to assiduously review the field of linear optical vortex arrays to identify novel aspects of their research that can be consolidated into a cornerstone for a much-revised manuscript and subsequently well-received addition to the field.

Reply: We thank the Reviewer for this highly pertinent suggestion. Previous studies of optical vortex arrays/lattices in linear systems mainly focus on methods for designing and generating desired vortex-antivortex structures, while neglecting the intrinsic couplings between vortices and antivortices. For example, the VAV arrays have been constructed by superimposing distinct light modes with engineered weighting coefficients. This has been demonstrated based on superpositions of the Hermite-Gauss beams [*Opt. Express* 19, 10293 (2011), Ref. [16]], Laguerre-Gauss beams [*Opt. Laser Eng.* 78, 132-139 (2016), Ref. [17]], Bessel-Gauss beams [*Phys. Rev. A* 108, 023509 (2023), Ref. [19]], Ince-Gauss beams [*Annalen der Physik* 533, 2000575 (2021), Ref. [18]] and perfect optical vortex (POV) beams [*Annalen der Physik* 529, 1700285 (2017), Ref. [20]; *Opt. Express* 28,

13775-13785 (2020), Ref. [72]]. Furthermore, multiple vortex modes including Laguerre-Gauss and POV modes with different transverse locations of vortex pivots can be combined into VAV arrays [*Opt. Express* 12, 1144-1149 (2004), Ref. [27]; *Opt. Lett.* 40, 2513-2516 (2015), Ref. [29]; *Opt. Express* 24, 28270-28278 (2016), Ref. [30]; *Appl. Phys. Lett.* 116, 011101 (2020), Ref. [31]; *Opt. Express* 30, 31959-31970 (2022), Ref. [32] etc.]. Taking into regard the diffraction of incident plane waves (or vortex beams) passing through an array of pin holes, the resultant interference distributions are able to form VAV lattices with square, hexagonal and honeycomb structures [*Appl. Opt.* 46, 2893-2898 (2007), Ref. [23]; *Opt. Commun.* 365, 99-102 (2016), Ref. [25]]. However, the propagation dynamics of these VAV arrays/lattices induced by the couplings between vortex-antivortex pivots was not systematically studied. The presence of the nonlocal couplings between vortices and antivortices makes them strongly unstable in the course of the propagation (an example of an unstable structure obtained by the above method is illustrated in Fig. S1).

There are recent works which have demonstrated the coupling between vortex and antivortex pivots, viz., Refs. [40], [42], and [43] in the main text. However, those works addressed the cases with simple VAV elements, rather the large-size VAV lattices. In particular, studied were a single elliptical vortex propagating in a linear medium (*Phys. Rev. A* 104, 043306 (2021), Ref. [36]), the coupling which keeps together the VAV pair (Refs. [34] and [36]), and re-creation of VAV pairs in a quadrupole structure, Ref. [37]. Unlike those single- and two-vortex settings, we report the large-scale crystallization of vortices and antivortices as a result of the balanced vortex-antivortex interaction. Specifically, we demonstrate, both theoretically and experimentally, a novel class of stable vortex-antivortex lattices supported by the intrinsic orbit-orbit coupling between neighbouring vortices nested in a freely propagating Gaussian light field.

Thus, the theoretical formalism and the concept of the optical vortex-antivortex crystallization, presented in this paper, are completely different from those in previous works. As recommended by the Reviewer, we have significantly revised the introductory section of the manuscript to clearly stress the novelty of the present work in comparison to the previous ones. The above-mentioned relevant references are cited in the revised manuscript.

As an aside, I would think that the very nice apparatus developed by the authors could be used to investigate arrays as analogs to two-dimensional condensed matter. For instance, phase diagrams could be developed to identify stable and unstable crystals associated with different arrangements of two-atom structures. Phase transitions could then be studied as well as order-disorder transformations. Polycrystalline materials could be constructed to observe and quantify grain boundary dynamics, allowing one to quantify surface tension at grain boundary surfaces and the kinetics of grain boundary triple points. Defects could be introduced and their kinetics compared, for instance, with defect structures in graphene lattices (<https://doi.org/10.1103/PhysRevLett.100.175503>). Leaning into the direction of optical matter seems to offer more promise than the vague nod made to optical communications that is often invoked by those more interested in underlying physics than the technological applications.

Reply: We thank the Reviewer for comments on these promising subjects. The comments

have greatly helped us to strengthen the paper by properly discussing these aspects. In particular, the observation of the VAV crystallization suggests the analogy to solid crystals, in addition to the fluidity demonstrated in previous works (such as *Phys. Rev. Lett.* 105, 163904 (2010) and *Nat. Commun.* 9, 2108 (2018), which are cited as Refs. [81] and [82], respectively, that illustrated effective fluidity of light). In this vein, the concept of “effective phase diagrams” can be put forward for describing phase transitions in the structured light. A recent work by A. Forbes (*Nat. Photonics* 16, 359-365 (2022), which is Ref. [33] in revised manuscript) demonstrates robust (self-healing) topological lattices in a cavity. As suggested by the present work, it may be very relevant to find more robust solidified optical structures. There is also a promising possibility to emulate other concepts of materials science by means of the structured light, such as the energy-band structure, entropy, and crystal defects (as mention by the Reviewer). On the other hand, in terms of potential applications, the optical lattices are very promising media in optical communications and all-optical data processing, as the lattices make it directly possible to extend the channel capacity. These subjects are properly discussed in the concluding section of the revised manuscript. The relevant papers mentioned above were cited as Refs. [33, 81-83].

Reviewer #2 (Remarks to the Author):

The authors claim to observe "Optical vortex-antivortex crystallization in free space" (free space meaning that it is not a nonlinear effect). If they really saw crystallization in free space, I would agree that this is a very interesting result – but they don't. They show that vortex-antivortex lattices are mostly stable and they call this "crystallization", but I think that "crystallization" implies a process in which a crystalline structure is formed. In contrast, the authors are simple showing that the structure is maintained.

Reply: We agree that this important point should be presented in a clearer form, although we do not quite agree with the conclusion made by the Reviewer. Stability is the most important characteristic of crystalline structures in real matter, where it is maintained by overall coherently balanced interaction of atoms (which corresponds, at the macroscopic level, to the interactions of vortex and antivortex pivots in optics). Naturally, the optical emulation of the crystallization should also satisfy the stability condition, i.e., the persistence of the VAV (vortex-antivortex) lattices. In this sense, our experimental results are indeed similar to their counterparts in condensed matter. On the other hand, there is an essential difference from the "genuine crystallization" (although not the same difference mentioned by the Review in his/her comment), because the solid-state crystallization occurs in very large lattices, while the experiment in optics is necessarily limited to relatively small patches of lattices. In fact, these similarities and differences were mentioned in the original manuscript. We have made an effort to stress them in a clearer form in the revised text. As concerns the “process in which a crystalline structure is formed”, this important point is directly addressed in our response to the next Reviewer’s comment.

It looks like the authors agree with my definition because in the abstract, they say "Here, we demonstrate that multiple optical VAV clusters nested in the propagating coherent field can crystallize into new patterns which preserve their lattice structure in the course of the linear propagation over distance up to several Rayleigh lengths." But in their data, they are

looking for a maintained shape – I don't see a single figure that starts with a random or disordered configuration and observes a lattice appearing with propagation (i.e. "crystallization").

Reply: We thank the Reviewer for this relevant comment. To address it, we have included essentially new experimental results in Fig. S2, showing a crystallization process of a 7×7 vortex-antivortex lattice. Figure S2(a) shows the vortex-antivortex pattern at a distance of $z=1.48z_R$, with a measured PCC value of 0.74 that indicates a nearly irregular lattice; while Fig. S2(b) presents a measurement showing a generated regular vortex-antivortex lattice from the irregular one. The measured PCC value increases to 0.96. Figures S2(c) and S2(d) show the corresponding vortex and antivortex pivot distributions to Fig. S2(a) and S2(b), respectively. The measurements directly demonstrate the crystallization process which transforms a *disordered VAV lattice* into a *regular* one. In a still more cogent form, such a process is also revealed by the vortex pivot trajectories in Fig. 5(a-c) and their transverse projection in Fig.5 (d-f) in the revised paper: vortices which are located at the edge and far away from the center gradually move towards positions which the lattice site should have in the regular lattice, and then they stay in those positions in the course of the subsequent long propagation. In the revised version, we have added details of these measurements to the supplement (Sec. E), and highlighted them in the main text, please see page 10.

Fig.S2: Additional measurements showing the vortex-antivortex crystallization. **a** and **b** depict the measured intensity distributions of the light fields at $z=-1.48z_R$ and 0; while **c** and **d** illustrate the corresponding distributions of vortex and antivortex pivots. The parameters used here are the same as those in Fig. 4 of the main text.

The only thing I see is Fig. 4g, which shows a very small increase in the "PCC" as the beam goes through a focus, but the increase is still within the error bars and the vortex

lattice structure is not discernibly different (there seems to be more rotation in the vortices further from the waist, but the vortex centers don't seem to be significantly further apart).

Reply: This comment is a relevant one too, calling for necessary clarification of the presentation of the material in the paper. PCC is an important indicator which identifies the stability and solidity of the VAV lattice. Taking results in Fig. S2 into account, we obtained the PCC values as $PCC=0.74$ and 0.96 at $z=-1.48 z_R$ and 0 , respectively, clearly demonstrating a developing crystallization process. Note that $PCC < 0.8$ indicates a disordered pattern. The stability of the VAV lattice is further characterized by the dynamical PCC curves in Fig. 5(g) in the revised manuscript, depicted for a shorter distance. We stress that solely the intensity and phase distributions are not sufficient to quantify the crystallization and stable propagation. This point is made clear in the revised paper, please see page 11, as well as Sec. E in the supplement.

Furthermore, it is unclear from Fig 2a where the imaging plane of the SLM is; if it is at $z=0$ (as I suspect), then it is trivial that the best PCC is here, since any defects will cause distortions away from the imaging plane.

Reply: We understand the comment, but disagree with it. The imaging plane is definitely the one of $z = 0$, in which the lattices are displayed. It is of course true that the light field is distorted in the course of the propagation. The nearly unaltered PCCs definitely corroborate the robustness of the VAV lattices and their straightforward similarity with solid-state lattices, hence these results are nontrivial. This point is well corroborated by our counterexamples, that use input configurations in which the polarity of the vortex pivots is (deliberately) inappropriately configured. Then, the respective PCC decays much quicker than observed in the stable lattices (please see Fig. 6g in the revised main text), implying the degradation of the input into a disordered speckle pattern (Fig. 6c in the revised manuscript). The motivation for using PCC and the interpretation of the observed results is properly clarified in the revised manuscript, please see page 11.

Other comments:

The second paragraph closes with "The instability spontaneously transforms initially regular lattices into disordered speckles [21–23]." I looked at these references, and it looks like these three papers do not deal with propagation dynamics of regular lattices. I don't see how any of them supports the claim that the authors make.

Reply: Those three works demonstrate that unbalanced VAV coupling quickly leads to complex screw motion (e.g., singularity scattering is demonstrated in Refs. [21,22], which are Refs. [43,44] in the revised manuscript), thus forming disordered speckle fields. This conclusion and Ref. [23] (which is Ref. [45] in the revised manuscript) are corollaries of Refs. [21,22] (Refs. [43,44] in the revised manuscript). We agree that this point was originally presented in the form which might be confusing. In the revised text, the respective statement is written, on page 2, as "The instability caused by the VAV interaction in Refs. [43-45] transforms initially regular lattices into disordered speckle patterns".

In contrast, this paper shows that the only way to get disordered speckle from a lattice is to disorder the lattice (so that it isn't a lattice anymore)

A Balbuena Ortega, S Bucio-Pacheco, S Lopez-Huidobro, L Perez-Garcia, FJ Poveda-Cuevas, JA Seman, AV Arzola, and K Volke-Sepúlveda. Creation of optical speckle by randomizing a vortex-lattice. *Optics express*, 27(4):4105– 4115, 2019.

Reply: We thank the Reviewer for drawing our attention to this relevant paper. Owing to the characteristic of isotropic diffraction in the free space, the totally disordered field is obtained with the help of random modulations (e.g., modulations of the scattering medium, and random vortex arrangement). The propagated pattern displayed by our counterexample in Fig. 6c of the main text at $z = 400$ mm is actually a quasi-disordered pattern, where four speckle-like segments are located symmetrically. This point is clarified in the revised manuscript, and we have added the reference mentioned by the Reviewer in an appropriate context (Ref. [46] in the revised main text).

At the end of the third paragraph, the authors state "Until now, stable VAV crystalline structures have not been reported in the linear propagation regime." In addition to the paper mentioned above, what about this paper?

Alexander Dreischuh, Sotir Chervenkov, Dragomir Neshev, Gerhard G Paulus, and Herbert Walther. Generation of lattice structures of optical vortices. *JOSA B*, 19(3):550–556, 2002.

Reply: We thank the Reviewer for indicating this reference. However, it features a critical difference from our work: in that paper, VAV lattices were formed and stabilized by the saturable nonlinearity, which induced a virtual grating with the periodic modulation of the refractive index. We have added the reference to that paper, with a proper comment (Ref. [38] in the revised main text).

Optical vortices often coincide with dark spots in a beam, but not always so. The authors should invest the time and equipment necessary to build an experimental means of measuring the phase, where the vortices can be directly measured and the VAV character can be observed. Surprisingly, Fig 3 shows multiple beamsplitters, which suggests that the authors are set up for interferometric phase measurements. I think that Nat Comm readers expect to see phase in current papers.

Reply: We thank the Reviewer for this very relevant suggestion. Following it, in the revised paper we have implemented the *phase-reconstruction method* according to Ref. [84], which is relevant in this context. The method has enabled us to retrieve the phase pattern, using the single interference between a reference plane wave and objective ones, thus avoiding the phase shifting operations. Dynamical phase distributions of VAV lattices in different planes are exhibited at Figs. 2(d,f), 3(e-h,m-p) and 4(e-h,m-p) in the revised manuscript. The detailed procedure of the phase measurements is presented in section "Material and method" of the revised text. In addition to that, a particular example is provided in Sec. A of the revised supplement.

REVIEWERS' COMMENTS

Reviewer #1 (Remarks to the Author):

The revised manuscript has satisfactorily addressed the issues raised in my initial review. Thanks for this nice contribution to the field.

Reviewer #2 (Remarks to the Author):

I appreciate the care the authors took in responding to the reviewer concerns. However, I still see their work as incremental, with no significant advances over previous work on vortex lattices in optics. While I disagree with their weak definition of crystallization, I would be okay with it if they include it in the manuscript -- but with that definition and the demonstration provided, I definitely don't think it warrants publication in Nat Comm.

Editors, Nature Communications
Re: MS No. NCOMMS-23-55028-T
"Optical vortex-antivortex crystallization in free space"
by Haolin Lin *et al.*

Reviewer #1 (Remarks to the Author):

The revised manuscript has satisfactorily addressed the issues raised in my initial review. Thanks for this nice contribution to the field.

Reply: We thank you again for your positive comment and support to our revised manuscript.

Reviewer #2 (Remarks to the Author):

I appreciate the care the authors took in responding to the reviewer concerns. However, I still see their work as incremental, with no significant advances over previous work on vortex lattices in optics. While I disagree with their weak definition of crystallization, I would be okay with it if they include it in the manuscript — but with that definition and the demonstration provided, I definitely don't think it warrants publication in Nat Comm.

Reply: We thank you again for your positive comments to our responses. But we respectfully disagree with your opinion that our work has no significant advances over previous work on vortex lattices. Let's explain to this issue again. Previous studies of optical vortex arrays/lattices in linear systems mainly focus on methods for designing and generating desired vortex-antivortex structures, while neglecting the intrinsic couplings between vortices and antivortices. We find that the presence of the nonlocal couplings between vortices and antivortices makes them strongly unstable in the course of the propagation. Here, for the first time, we exploit the effect of balanced vortex-antivortex coupling and achieve both in theory and in experiment crystallization of the vortex-antivortex lattices. The presented vortex-antivortex crystallization which takes place in free space diametrically opposite to stable lattices produced by nonlinearities.

Two key characteristics featuring the crystallize vortex-antivortex structure have been clearly demonstrated. First, we have observed that a disordered vortex-antivortex lattices gradually transforms into a regular pattern in the course of propagation. Second, we have demonstrated that the crystalized regular pattern preserves its lattice structure over distance up to many Rayleigh lengths, featuring stable propagation. In the revised manuscript, we have additionally clarified our definition about the vortex-antivortex crystallization, as this reviewer suggested.